# Beyond Motor Deficits: Environmental Enrichment Mitigates Huntington’s Disease Effects in YAC128 Mice

**DOI:** 10.3390/ijms241612607

**Published:** 2023-08-09

**Authors:** Evelini Plácido, Priscilla Gomes Welter, Ana Wink, Gabriela Duarte Karasiak, Tiago Fleming Outeiro, Alcir Luiz Dafre, Joana Gil-Mohapel, Patricia S. Brocardo

**Affiliations:** 1Neuroscience Graduate Program, Center of Biological Sciences, Federal University of Santa Catarina, Florianopolis 88040-900, SC, Brazilaclaudiawink@gmail.com (A.W.); alcir.dafre@ufsc.br (A.L.D.); 2Department of Biochemistry, Center of Biological Sciences, Federal University of Santa Catarina, Florianopolis 88040-900, SC, Brazil; gabriela.karasiak@ufsc.br; 3Department of Experimental Neurodegeneration, Center for Biostructural Imaging of Neurodegeneration, University Medical Center Göttingen, 37075 Göttingen, Germany; touteir@gwdg.de; 4Max Planck Institute for Natural Sciences, 37075 Göttingen, Germany; 5Translational and Clinical Research Institute, Faculty of Medical Sciences, Newcastle University, Framlington Place, Newcastle Upon Tyne NE1 7RU, UK; 6Deutsches Zentrum für Neurodegenerative Erkrankungen (DZNE), 18147 Göttingen, Germany; 7Island Medical Program, Faculty of Medicine, University of British Columbia and Division of Medical Sciences, University of Victoria, Victoria, BC V8P 5C2, Canada; 8Department of Morphological Sciences, Center of Biological Sciences, Federal University of Santa Catarina, Florianopolis 88040-900, SC, Brazil

**Keywords:** environmental enrichment, Huntington’s disease, monoamines, neurodegeneration, neuronal plasticity, transgenic animal model, YAC128 mice

## Abstract

Huntington’s disease (HD) is a neurodegenerative genetic disorder characterized by motor, psychiatric, cognitive, and peripheral symptoms without effective therapy. Evidence suggests that lifestyle factors can modulate disease onset and progression, and environmental enrichment (EE) has emerged as a potential approach to mitigate the progression and severity of neurodegenerative processes. Wild-type (WT) and yeast artificial chromosome (YAC) 128 mice were exposed to different EE conditions. Animals from cohort 1 were exposed to EE between postnatal days 21 and 60, and animals from cohort 2 were exposed to EE between postnatal days 60 and 120. Motor and non-motor behavioral tests were employed to evaluate the effects of EE on HD progression. Monoamine levels, hippocampal cell proliferation, neuronal differentiation, and dendritic arborization were also assessed. Here we show that EE had an antidepressant-like effect and slowed the progression of motor deficits in HD mice. It also reduced monoamine levels, which correlated with better motor performance, particularly in the striatum. EE also modulated neuronal differentiation in the YAC128 hippocampus. These results confirm that EE can impact behavior, hippocampal neuroplasticity, and monoamine levels in YAC128 mice, suggesting this could be a therapeutic strategy to modulate neuroplasticity deficits in HD. However, further research is needed to fully understand EE’s mechanisms and long-term effects as an adjuvant therapy for this debilitating condition.

## 1. Introduction

Huntington’s disease (HD) is a hereditary, neurodegenerative disease that affects motor, cognitive, and neuropsychiatric functions [1,2]. The genetic cause of HD is an expanded CAG repeat in the *HD* gene located in the 4p16.3 position. This mutated gene confers an expanded polyglutamine stretch to the protein huntingtin (Htt) [3], which is widely expressed both within and outside of the central nervous system (CNS) [4]. However, despite the genetic cause of the disease being known since 1993 [5], no effective treatment can prevent or slow disease progression and severity [6,7].

Motor symptoms, such as chorea and loss of motor coordination, strongly correlate with severe neuronal loss in the striatum, the main neuropathological hallmark of the disease [8] and a commonly used marker of disease onset [2]. Despite the relative striatal selectivity, dysfunction, and degeneration in other regions of the brain, particularly the cerebral cortex and the hippocampus, are thought to underlie the neuropsychiatric symptoms and cognitive decline seen in HD individuals [9,10,11]. HD is also characterized by non-central features, such as weight loss, skeletal muscle wasting, and cardiac failure [12,13]. Of note, body weight has been shown to impact the rate of disease progression, with a higher body mass index being associated with a slower progression of the disease [14].

Alterations in neurotransmitter levels are also found in HD. Some studies have shown the upregulation of dopamine and norepinephrine levels in the basal ganglia [15,16]. In contrast, others showed a reduction in the striatal concentration of dopamine and its metabolite, homovanillic acid [17,18]. These differences have been attributed to different stages of disease progression [19,20]. Of note, tetrabenazine and deutetrabenazine, which have been approved by the U.S. Food and Drug Administration (FDA) to manage HD symptoms (mainly chorea), act as inhibitors of the vesicular monoamine transporter 2 (VMAT2), decreasing monoamine release in the synaptic cleft [21,22]. However, these drugs also increase the risk of depression and suicidal thoughts [22,23], already present in many HD patients [7].

HD is typically an adult-onset disease, although juvenile and late-onset cases also exist. The main contributing factor for the age of onset is the CAG repeat length, with longer expansions hastening disease onset [24,25]. However, intriguingly, individuals sharing identical CAG repeat lengths present variability in symptoms and age of onset [26,27]. Indeed, while the number of Htt CAG repeats can explain approximately 67% of the variability in age at onset [25], the remaining variability is attributed to genes other than the *HD* gene (40%) and environmental factors (60%) [26]. One significant environmental factor that has drawn attention is environmental enrichment (EE). EE encompasses factors that reinforce sensory, cognitive, motor, and social stimulation [28,29]. Studies have revealed that EE exposure can lead to increased hippocampal neurons [30] and BDNF levels [31] in animal models of HD. These effects can potentially have a therapeutic impact since cell loss, hippocampal dysfunction [9,32], and BDNF expression and intracellular transport are decreased in HD patients [31].

Given the potential beneficial impact that EE may have in HD, this study aimed to assess the effects of early exposure to an enriched environment on the development of central and peripheral alterations in the yeast artificial chromosome (YAC) 128 transgenic mouse model of HD.

## 2. Results

### 2.1. Body Weight

A two-way ANOVA revealed a significant effect of genotype [F(1,34) = 5.54; *p* = 0.02] but no effect of housing environment [F(1,34) = 0.88; *p* = 0.35] and no genotype versus environment interaction [F(1,34) = 3.46; *p* = 0.07] on body weight at two months of age. Further post hoc analyses revealed that YAC128 mice exhibited a significant weight gain (*p* = 0.02) compared to their age-matched WT counterparts in the CE. In contrast, this genotypic difference was not found in EE (*p* > 0.99). At four months of age, there were no significant differences in body weight among the groups (two-way ANOVA: genotype [F(1,32) = 2.66; *p* = 0.11], environment [F(1,32) = 0.09; *p* = 0.77], genotype versus environment interaction [F(1,32) = 2.20; *p* = 0.15]) (Table 1).

### 2.2. Effects of Genotype and Environment on Motor Performance

To determine the effect of genotype and EE on motor coordination and balance, the performance of 2- and 4-month-old YAC128 mice and their age-matched WT controls was assessed using the rotarod test. Two-month-old YAC128 mice showed a reduced latency time to fall (*p* = 0.02) as compared to WT (Figure 1A), and EE exposure was able to reverse this deficit (*p* = 0.002) (two-way ANOVA: genotype [F(1,34) = 4.79; *p* = 0.03]; environment [F(1,34) = 14.04; *p* = 0.001]; genotype versus environment interaction [F(1,34) = 4.76; *p* = 0.04]) (Figure 2A). In addition, the number of falls was also reduced in YAC128 mice exposed to EE (*p* = 0.007) (two-way ANOVA: genotype [F(1,33) = 4.02; *p* = 0.05]; environment [F(1,33) = 13.59; *p* = 0.001]; genotype versus environment interaction [F(1,33) = 2.07; *p* = 0.16]) (Figure 1B).

In 4-month-old animals, there was an effect of the interaction between genotype and environment [F(1,48) = 4.10; *p* = 0.05]; however, there was no effect of the genotype [F(1,48) = 2.22; *p* = 0.14] or environment [F(1,48) = 0.23; *p* = 0.63] on the latency to fall (Figure 1C). A difference in the number of falls was detected between YAC128 and WT mice [F(1,48) = 10.08; *p* = 0.003], but no effect of environment [F(1,48) = 0.80; *p* = 0.38] or genotype versus environment interaction [F(1,48) = 2.43; *p* = 0.12] was found (Figure 1D).

The pole test was also performed to evaluate the effects of genotype and EE on motor coordination. In 2-month-old animals, a two-way ANOVA did not reveal significant genotype effects [F(1,33) = 0.08; *p* = 0.77], environment [F(1,33) = 0.06; *p* = 0.81] effects, or a significant environment versus genotype interaction [F(1,33) = 2.33; *p* = 0.14] with regard to the total distance covered in the OFT. Similar findings were obtained in 4-month-old animals (two-way ANOVA: genotype [F(1,48) = 1.94; *p* = 0.17], environment [F(1,48) = 1.71; *p* = 0.20], environment versus genotype interaction [F(1,48) = 0.91; *p* = 0.34]) (Appendix A: Behavioral results for YAC128 and WT mice at 2 and 4 months of age).

### 2.3. Effects of Genotype and Environment on Locomotion and Anxious-like Behavior

The OFT was performed to evaluate the effects of genotype and EE on locomotion and anxious-like behavior. In 2-month-old animals, a two-way ANOVA revealed no significant effects of genotype [F(1,33) = 0.08; *p* = 0.77] and environment [F(1,33) = 1.71; *p* = 0.20] and no significant environment versus genotype interaction [F(1,33) = 0.62; *p* = 0.44] with regard to the total distance covered in the OFT. Similar findings were also obtained regarding time spent in the center of the apparatus (two-way ANOVA: genotype [F(1,33) = 0.02; *p* = 0.87], environment [F(1,33) = 0.26; *p* = 0.61], environment versus genotype interaction [F(1,33) = 0.66; *p* = 0.42]) (Appendix A: Behavioral results for YAC128 and WT mice at 2 and 4 months of age).

Similarly, at 4 months of age, an ANOVA also revealed no significant effects of genotype [F(1,32) = 0.03; *p* = 0.86] and environment [F(1,32) = 0.89; *p* = 0.35] and no significant environment versus genotype interaction [F(1,32) = 2.80; *p* = 0.10] with regard to the total distance covered in the OFT. Similar findings were also obtained concerning time spent in the center of the apparatus (two-way ANOVA: genotype [F(1,32) = 1.87; *p* = 0.18], environment [F(1,32) = 0.78; *p* = 0.38], environment versus genotype interaction [F(1,32) = 1.29; *p* = 0.26]) (Appendix A: Behavioral results for YAC128 and WT mice at 2 and 4 months of age).

### 2.4. Effects of Genotype and Environment on Depressive-like Behavior

To assess the effects of EE on depressive-like behaviors, YAC128 mice and their age-matched WT counterparts were subjected to the TST at 2 and 4 months of age. A two-way ANOVA was performed, and no significant effect of the genotype [F(1,27) = 2.71; *p* = 0.11] and environment [F(1,27) = 0.50; *p* = 0.48], as well as no significant genotype versus environment interaction [F(1,27) = 1.35; *p* = 0.25], was found with regard to the immobility time in the TST (Figure 2A) at 2 months of age. In 4-month-old animals, significant effects of genotype [F(1,31) = 11.47; *p* = 0.002] and environment [F(1,31) = 14.57; *p* = 0.001] were detected, although no significant interaction between genotype and environment was observed [F(1,31) = 3.17; *p* = 0.08]. Further, post hoc analyses revealed that YAC128 mice remained immobile longer than WT animals exposed to CE (*p* = 0.004), and EE prevented this deficit in YAC128 mice (*p* < 0.001) (Figure 2B).

The ST was further used to assess self-care and motivational behavior. A two-way ANOVA revealed a significant effect of the environment [F(1,33) = 4.474; *p* = 0.04] but no significant effect of genotype [F(1,33) = 0.002; *p* = 0.97] or significant genotype versus environment interaction [F(1,33) = 0.062; *p* = 0.80] on the latency to grooming (Figure 2C) in 2-month-old animals. The time spent grooming revealed an effect of the environment [F(1,32) = 5.586; *p* = 0.02], but no effect of genotype F(1,32) = 1.793; *p* = 0.19] or genotype versus environment interaction [F(1,32) = 0.481; *p* = 0.49] at this time-point (Figure 3E). At four months of age, a two-way ANOVA showed a significant effect of genotype [F(1,31) = 4.27; *p* = 0.05) but no significant effect of environment [F(1,31) = 0.15; *p* = 0.86] or significant genotype versus environment interaction [F(1,31) = 0.03; *p* = 0.86] on the latency to grooming. As for the time spent grooming, a significant effect of genotype [F(1,31) = 4.47; *p* = 0.04), but no significant effect of environment [F(1,31) = 0.39; *p* = 0.53] and no significant genotype versus environment interaction [F(1,31) = 2.21; *p* = 0.15] were detected. However, further post hoc analysis showed no significant differences among groups (Figure 2D,F).

### 2.5. Effects of Genotype and Environment on Monoamine Levels

Monoamine levels in the striatum, hippocampus, and prefrontal cortex were measured by HPLC using brain samples from WT and YAC128 mice at 2 months of age.

*Striatum:* A two-way ANOVA revealed an effect of environment on striatal levels of 5-HT [F(1,33) = 9.69; *p* = 0.004], NE [F(1,31) = 7.79; *p* = 0.01], and DA [F(1,31) = 11.83; *p* = 0.002] in 2-month-old mice. Further post hoc testing revealed that EE decreased the levels of NE in YAC128 mice (*p* = 0.03) (Figure 3B) as well as the levels of DA in WT mice (*p* = 0.04) (Figure 3C).

*Hippocampus:* A two-way ANOVA revealed no significant effects of genotype [F(1,32) = 0.002; *p* = 0.96] and environment [F(1,32) = 0.14; *p* = 0.71] and no significant genotype versus environment interaction [F(1,32) = 0.88; *p* = 0.35] with regard to hippocampal 5-HT levels (Figure 3D). However, there was a significant effect of environment in the hippocampal levels of NE [F(1,32) = 7.45; *p* = 0.01], but no significant effect of genotype [F(1,32) = 0.002; *p* = 0.96], and no significant genotype versus environment interaction [F(1,32) = 0.14; *p* = 0.71] was found (Figure 3E).

*Prefrontal Cortex:* There was also a significant effect of environment in the cortical levels of 5-HT [F(1,34) = 4.11; *p* = 0.05] but no significant effect of genotype [F(1,34) = 0.27; *p* = 0.60] and no significant genotype versus environment interaction [F(1,34) = 0.62; *p* = 0.44] (Figure 3F). About cortical NE levels, there was a significant genotype versus environment interaction [F(1,33) = 4.12; *p* = 0.05] and a significant effect of genotype [F(1,33) = 0.38; *p* = 0.05] but no significant effect of the environment [F(1,33) = 1.78; *p* = 0.19].

A correlation analysis between rotarod performance and monoamine levels was conducted, and significant negative correlations were found between latency to fall in the rotarod and striatal 5-HT (*p* = 0.01; Figure 4A), striatal NE (*p* = 0.003; Figure 4B), and hippocampal NE (*p* = 0.03; Figure 5E) levels. Therefore, the results indicate a possible association between lower levels of these monoamines and better motor performance.

The levels of monoamines (DA, NA, and 5-HT) were also evaluated in the hippocampus, prefrontal cortex, and striatum of 4-month-old animals (Figure 5).

*Striatum:* No significant effects were found with regard to the levels of 5-HT (two-way ANOVA: genotype [F(1,15) = 0.38; *p* = 0.55]; environment [F(1,15) = 1.34; *p* = 0.26]; genotype versus environment interaction [F(1,15) = 0.29; *p* = 0.60]), NA (two-way ANOVA: genotype [F(1,15) = 0.43; *p* = 0.52]; environment [F(1,15) = 1.30; *p* = 0.27], genotype versus environment interaction [F(1,15) = 3.72; *p* = 0.07]), and DA (two-way ANOVA: genotype [F(1,15) = 2.07; *p* = 0.17]; environment [F(1,15) = 1.08; *p* = 0.32], genotype versus environment interaction [F(1,15) = 0.08; *p* = 0.78]) in the striatum of 4-month-old mice.

*Hippocampus:* Similarly, no significant effects were found with regard to the levels of 5-HT (two-way ANOVA: genotype [F(1,15) = 0.01; *p* = 0.93]; environment [F(1,15) = 1.64; *p* = 0.22], genotype versus environment interaction [F(1,15) = 1.40; *p* = 0.25]) and NA (two-way ANOVA: genotype [F(1,15) = 0.01; *p* = 0.91]; environment [F(1,15) = 1.76; *p* = 0.20], genotype versus environment interaction [F(1,15) = 0.54; *p* = 0.47]) in the hippocampus of 4-month-old mice.

*Prefrontal Cortex:* Again, no significant effects were found with regard to the levels of 5-HT (two-way ANOVA: genotype [F(1,15) = 0.23; *p* = 0.64]; environment [F(1,15) = 0.28; *p* = 0.61], genotype versus environment interaction [F(1,15) = 3.05; *p* = 0.10]) and NA (two-way ANOVA: genotype [F(1,15) = 0.11; *p* = 0.74]; environment [F(1,15) = 2.21; *p* = 0.16], genotype versus environment interaction [F(1,15) = 0.33; *p* = 0.57]) in the prefrontal cortex of 4-month-old mice.

### 2.6. Effects of Genotype and Environment on Hippocampal Cell Proliferation

To analyze the potential effects of EE on DG cell proliferation in YAC128 mice, we used the endogenous cell proliferation marker Ki-67 [33]. At 2 months of age, a two-way ANOVA revealed no significant effects of genotype [F(1,12) = 0.26; *p* = 0.62] and environment [F(1,12) = 0.66; *p* = 0.43] but a significant genotype versus environment interaction [F(1,12) = 6.56; *p* = 0.02] (Figure 6E). In 4-month-old animals, a two-way ANOVA also revealed a significant genotype versus environment interaction [F(1,20) = 16.29; *p* < 0.0001] and no significant effects of genotype [F(1,20) = 0.29; *p* = 0.59] or environment [F(1,20) = 0.00; *p* = 0.99]. Additional post hoc analysis detected a significant increase in proliferating cells in the DG subgranular zone (SGZ) of YAC128 mice housed in CE compared to their WT littermate controls (*p* = 0.02). However, this difference was not found when the animals were exposed to EE (*p* = 0.13) (Figure 6F).

### 2.7. Effects of Genotype and Environment on Hippocampal Neuronal Differentiation and Dendritic Arborization

Since we found a significant effect of the EE with regard to cell proliferation in 4-month-old animals, we subsequently used the endogenous neuronal differentiation marker DCX, a microtubule-binding protein expressed in newly differentiated and migrating neuroblasts [34], to investigate the effects of EE on neuronal differentiation. A two-way ANOVA revealed significant effects of genotype [F(1,19) = 8.62; *p* = 0.01] and environment [F(1,19) = 48.34; *p* < 0.0001] and a significant genotype versus environment interaction [F(1,19) = 8.53; *p* = 0.01] with regard to DG neuronal differentiation. Further post hoc analysis revealed that EE significantly increased the number of DCX-positive cells in the total DG of WT mice (*p* < 0.0001) (Figure 7I).

To further characterize this finding, we also calculated the total number of DCX-positive cells in the dorsal and ventral portions of the DG. In the dorsal DG, a two-way ANOVA revealed significant effects of genotype [F(1,18) = 9.06; *p* = 0.01] and environment [F(1,18) = 39.19; *p* < 0.0001], as well as a significant genotype versus environment interaction [F(1,18) = 7.84; *p* = 0.01] with regard to the number of DCX-positive cells. Further post hoc analyses demonstrated that EE significantly increased the number of DCX-positive cells in WT mice (*p* < 0.0001). However, YAC128 mice exposed to EE showed a significantly lower number of DCX-positive cells than WT mice exposed to EE (*p* = 0.002) (Figure 7J). In the ventral DG, a two-way ANOVA revealed a significant effect of the environment [F(1,19) = 51.48; *p* < 0.0001] but no significant effect of genotype [F(1,19) = 2.71; *p* = 0.12] or genotype versus environment interaction [F(1,19) = 2.44; *p* = 0.13]. Post hoc analysis revealed that EE increased the number of DCX-positive cells in both genotypes (Figure 7K).

To evaluate the effect of EE on dendritic branching in WT and YAC128 DG immature neurons, Sholl analysis was performed on samples immunolabeled for the DCX protein (Figure 7L). The following characteristics were analyzed: maximum distance from the soma to the end of the dendrites and number of intersections per radius (Figure 7N). A two-way ANOVA revealed a significant effect of the environment [F(1,20) = 10.82; *p* = 0.004] but no significant effect of genotype [F(1,20) = 0.34; *p* = 0.57] and no significant genotype versus environment interaction [F(1,20) = 0.72; *p* = 0.41] with regard to the maximum distance from the soma to the end of the dendrites in DG immature neurons. Further post hoc analysis revealed that EE significantly increased this distance in dendrites from WT DG immature neurons (*p* = 0.05) (Figure 7M).

The complexity of the dendritic arborization of DG immature neurons was evaluated by counting the number of intersections per radius through a Sholl analysis. Repeated-measure ANOVA revealed a significant effect of the environment [F(126,2520) = 4.92; *p* < 0.01] and a significant genotype versus environment interaction [F(126,2520) = 1.52; *p* < 0.01], but no significant effect of genotype [F(126,2520) = 0.24; *p* = 1.00] with regard to the number of intersections per radius (Figure 7N).

## 3. Discussion

HD is an inherited neurological condition classically associated with movement disturbances [35]. Despite that, movement impairments are not the only symptom of HD [36]. Indeed, in a recent study including participants with manifest and premanifest HD, the three most prevalent symptoms reported were emotional issues, fatigue, and difficulty thinking [36]. Given this rationale, in the present study, we evaluated both motor and non-motor symptoms in the YAC128 HD transgenic mouse model exposed to EE.

Among the genetic models used to study HD, YAC128 is considered an excellent model due to its face validity (YAC128 mice mimic several of the clinical symptoms of HD, such as motor deficits), construct validity (YAC128 mice present pathophysiological characteristics of HD, such as striatal neuronal death), and predictive validity (YAC128 mice respond to classic clinical treatment used to treat their motor symptoms of HD) [37,38,39,40]. In addition, YAC128 mice show a slow progression of symptoms similar to humans [41].

### 3.1. Effects of EE Exposure on Non-Motor Symptoms in YAC128 Mice

Alterations in body weight are seen in both HD patients and YAC128 HD mice, albeit in opposite directions. Indeed, while weight loss is reported in the early and later clinical stages of HD [42,43], YAC128 mice show weight gain [44]. BACHD mice, another full-length transgenic model of HD, also show weight gain [44], whereas R6/2 mice, a truncated HD model, show weight loss [45]. The findings in this study agree with the literature [46] since, in this study, YAC128 mice also showed body weight gain as early as 2 months of age. Furthermore, EE prevented this gain, in agreement with previous studies [47]. Of note, although a previous study showed that YAC128 mice were heavier than their littermates at 3, 6, 9, and 12 months of age [48], we did not find significant weight differences in 4-month-old YAC128 mice in the present study.

Another relevant aspect known to impact the lives of HD patients early on in the course of the disease is the occurrence of psychiatric symptoms, including depression [11,49,50,51,52,53]. These can be present in HD-afflicted individuals’ years before the onset of motor symptoms [36,54]. In agreement with our previous study [55], in the present study, we only found depressive-like behaviors in YAC128 mice at 4 months of age but not at 2 months of age. Notably, EE prevented the occurrence of depressive-like behaviors in 4-month-old YAC128 mice. Importantly, these effects cannot be attributable to locomotor alterations since EE exposure significantly decreased immobility time in the TST without altering locomotor activity in the open field test (Appendix A: Behavioral results for YAC128 and WT mice at 2 and 4 months of age).

The open field test was also used to evaluate anxiety-like behaviors, constituting another significant complaint for HD patients [11,56]. Here we did not find a significant difference in the time spent in the center of the open field at either 2 or 4 month of age (Appendix A: Behavioral results for YAC128 and WT mice at 2 and 4 months of age). Although our previous report had found genotypic differences in anxiety-like behaviors at 2 months of age (likely due to litter variability), no differences at 4 months were also detected in our previous study [55].

### 3.2. Effects of EE Exposure on Motor Symptoms in YAC128 Mice

In HD transgenic mouse models, a test frequently used to assess motor deficits is the rotarod test. Motor performance on the rotarod can be influenced by several factors, such as motor coordination, balance, motor learning, and fatigue [57,58,59,60]. To some degree, all of these features can be present in HD [36,61,62,63]. The acceleration protocol applied in this study sought to analyze the combined effect of these various factors on motor performance, thus differing from other protocols that attempt to separate one of the contributing factors to motor impairment (for example, a fixed and slow speed minimizes the influence of cardiopulmonary performance/fatigue; [57]). In the present study, we found impairments in the accelerating rotarod test regarding latency to fall, the number of falls at 2 months, and the number of falls at 4 months of age. Of note, EE was able to improve motor performance in YAC128. Previous studies have shown that although both WT and YAC128 mice can learn the rotarod test, YAC128 mice require more training to reach the same level of performance as WT mice at 2 months of age [64,65]. Moreover, impaired performance in the acceleration protocol was found in 3-month-old YAC128 mice [66], although this has not been reported at this age in other studies [48]. Our study supports these findings and indicates that motor disturbances can be seen in YAC128 mice as early as 2 months of age with an accelerating rotarod protocol.

### 3.3. Effects of EE Exposure on Monoamine Neurotransmission in YAC128 Mice

Although HD is a neurodegenerative disease, many symptoms manifest before significant neurodegeneration [8], arguing against neuronal cell death being the direct cause of the observed symptoms. Indeed, there are reported cases with clinically recognizable HD but without gross or microscopic abnormalities [67], suggesting the presence of alternative pathological alterations. The same occurs in the YAC128 HD mouse model. While YAC128 mice present physical, cognitive, mood, and motor alterations before 4 months, neurodegeneration only emerges at older ages [46,48,68]. On the other hand, neuropathological alterations other than neuronal death can be seen in YAC128 brains at different ages. For example, analysis of myelinated fibers in the corpus callosum indicates that myelin sheaths are thinner in YAC128 mice as early as 1.5 months [69]. Another study found a significant reduction in extracellular DA concentrations in the striatum as assessed by in vivo microdialysis in 7-month-old YAC128 mice [68]. In the present study, we investigated the effects of genotype and EE on monoamine levels, cell proliferation, neuronal differentiation, and neuronal arborization.

We found that EE exposure decreased striatal NE and DA levels, consistent with the mechanism of action of drugs used to treat chorea in HD. Interestingly, the EDD-induced decrease in NE in the striatum was significantly correlated with better rotarod performance. In addition, better rotarod performance was also correlated with lower levels of 5-HT in the striatum and NE in the hippocampus. Although no genotype effects on monoamine levels were found in this study at either 2 or 4 months of age, previous studies have found alterations in monoamine levels in both HD patients and HD animal models [20,70]. These contradictory results are probably related to the age of the animals and individuals and the stages of disease progression across the different studies [19,20]. For example, DA and NE concentrations were elevated in the basal ganglia of HD carriers [15,16], while in others, the concentration of DA in the striatum was reported to be reduced [17,18]. Additionally, a study involving HD patients showed increased DA levels but no alteration in NE levels in the cerebrospinal fluid study of HD patients [71]. The influence of age on reported discrepancies in monoamine levels is further supported by findings that homovanillic acid (HVA), a metabolite of DA, was higher in symptomatic HD patients but reduced in premanifest HD individuals [72]. In preclinical studies, a reduction in striatal levels of NE and DA has been reported in 5-month-old YAC128 mice [70], while a decrease in mRNA levels of striatal D1 and D2 DA receptors was found in 10-month-old YAC128 mice [65]. Evidence of dysregulation of neurotransmitters released in the hippocampus, assessed by electrophysiological measurements, has also been shown in early-symptomatic YAC128 mice [73]. It is important to note that NE is synthesized from DA by dopamine β-hydroxylase, and there is evidence that DA and NE can act in an overlapping manner [16,74].

In the present study, we found that EE could improve motor performance, possibly due, at least in part, to a concomitant decrease in monoamine levels. Besides the negative correlation of 5-HT and NE in the striatum with better performance in rotarod evaluation, the levels of NE in the hippocampus were also significantly correlated with better motor performance. Although it has been proposed that the hippocampus is not directly related to the motor component of the rotarod task, including motor learning [75,76,77], a previous study has found a correlation between rotarod performance and fractional anisotropy (FA) derived from diffusion in the hippocampus as assessed with MRI, suggesting its potential role in motor tasks [77].

### 3.4. Effects of EE Exposure on Neuronal Plasticity in YAC128 Mice

Adult hippocampal neurogenesis has been thought to contribute, at least in part, to cognitive and affective functions [78,79], and deficits in the neurogenic process have been proposed to contribute to the cognitive disturbances and changes in affective behaviors seen in HD [80]. Many studies have described that EE increases the genesis of new neurons and their survival in the hippocampal DG [30,81,82]. In the present study, we found that EE exposure increases hippocampal cell proliferation (Ki-67-positive cells) and neuronal differentiation, as the number of DCX-positive cells was increased by EE exposure across the entire dorsal–ventral axis of the hippocampus. Of note, this effect could be mediated, at least in part, by providing EE animals with access to a running wheel, as physical exercise is known to exert a powerful antidepressant effect [83] and to have powerful pro-neurogenic properties [82].

Of note, EE can also induce changes in the nervous system by promoting the reorganization of the cytoskeleton in neurons. In rodents, EE has been shown to increase dendritic arborization [84,85], whereas dendritic morphological abnormalities have been observed in both HD patients and in animal models of HD [86]. Previous studies have proven that DCX is an effective marker for dendritic arborization in immature neurons [87]. In this study, we used Sholl analysis [88] to assess the effect of EE on dendritic arborization in DG immature neurons. We found that the maximum distance reached by the dendrites was significantly greater in WT mice exposed to EE. However, we did not find significant differences regarding dendritic arborization between WT and YAC128 mice. These results contrast those reported by the authors of [89], who demonstrated that EE could increase dendritic arborization in WT mice and the R6/1 HD mouse model. The discrepancy between our results and those reported in [89] may be due to differences between the HD transgenic mouse models used in the two studies (i.e., between the R6/1 and the YAC128 models).

Although previous studies have shown that “cognitive enrichment” can improve motor performance in the YAC128 mouse model (while interestingly not impacting cognitive performance) [90], to the best of our knowledge, this is the first time that the effect of EE in the home cage was evaluated in the YAC128 HD mouse model. Furthermore, it has been previously reported that monoamine levels are altered in YAC128 mice at 5 months of age [70]; the present study is also the first to assess the effect of EE on monoamine levels in YAC128 mice. Our results indicate that EE can improve motor performance, influence monoamine levels and neuronal differentiation, and positively affect affective-related behaviors. These findings suggest that the monoamine system may mediate, at least in part, the beneficial effects associated with EE, as supported by the significant correlation reported in this study.

## 4. Material and Methods

### 4.1. Animals

Male and female YAC128 transgenic and wild-type (WT) mice (line 53, FVB/N background strain) were used for these experiments (detailed descriptions of these transgenic animals can be found elsewhere; [48]). All animals were generated from our local colony with breeding couples generously provided by Dr. Brian Christie (University of Victoria, Canada). Animals were weaned and separated according to sex at postnatal day (PND) 21 and divided into two groups. Mice were maintained in a standard 12 h light/dark cycle (lights on at 0700 h) and at 20–22 °C with free access to water and food. All manipulations were carried out during the light period. All experimental procedures were conducted following the National Institutes of Health Guide for the Care and Use of Laboratory Animals and were approved by the Committee on Ethics of Animal Experimentation of the Federal University of Santa Catarina (Florianópolis, Brazil; Protocol Number: 4502210318). All efforts were made to minimize animal suffering and the number of animals used in these experiments.

### 4.2. Genotyping

DNA extracted from mouse ear tissue using Wizard Genomic DNA Purification Kit (Promega, Madison, WI, USA) was used for genotyping animals by polymerase chain reaction (PCR), using primers for detection of YAC LYA (LYA1 = 5′ CCTGCTCGCTTCGCTACTTGGAGC 3′, LYA2 = 5′ GTCTTGCGCCTTAAACCAACTTGG 3′) and RYA (RYA2 = 5′ CCGCACCTGTGGCGCCGGTGATGC 3), as previously described by us [91]. Polymerase chain reaction (PCR) products were run on a 1.5% agarose gel with 5000× SYBR-safe nucleic acid gel stain (Invitrogen, Carlsbad, CA, USA), using a GeneRuler 100 bp DNA Ladder (Thermo Scientific, Waltham, MA, USA) and visualized under a ChemiDoc Imager (Bio-Rad Laboratories, Hercules, CA, USA).

### 4.3. Experimental Procedure

Cohort 1: On postnatal day (PND) 21, YAC128 and WT mice were weaned and immediately housed in standard housing (control environment, CE) or an enriched environment (EE) in groups of approximately 6 animals per cage (Figure 8). CE animals were housed in a standard cage (29 × 18 × 12 cm), whereas EE animals were housed in a larger cage (41 × 34 × 16 cm) containing four objects. Objects included plastic and cardboard tubes, twine, wire, nest boxes, hammocks made from surgical face masks, cotton for nesting, and Lego pieces. Running wheels were not used in cohort 1. These objects were changed twice weekly, with two remaining in the box and two being exchanged for different ones. Animals were housed in their respective environments for 40 consecutive days, from PND 21 to PND 60. At two months, animals underwent behavioral testing for four consecutive days. The battery of behavioral tests included: pole test (PT), open field test (OFT), splash test (ST), tail suspension test (TST), and rotarod test. All tests were performed during the light cycle under low-intensity light (12 lux). Twenty-four hours after the last behavioral test, animals were weighed, a subgroup of animals was euthanized by cervical dislocation, and the hippocampus, prefrontal cortex, and striatum were dissected to measure monoamine levels. Another subgroup of animals was euthanized by cardiac perfusion, and their brains were collected and used for immunohistochemistry.

Cohort 2: At PND 60, YAC128 and WT mice in the EE group were housed in groups of 4 to 6 animals in plastic cages (41 × 34 × 16 cm) enriched with objects similar to those described for cohort 1, except for running wheels being used for the entire duration of the experiment in this cohort. Novel objects were introduced 1 to 2 times per week. Mice in the CE group were housed in groups of 6 in standard opaque plastic cages (41 × 34 × 16 cm). They were left undisturbed for the entire experiment duration, except for cage cleaning and body weight measurements, performed weekly from weaning and throughout the experimental period (this was similar for all other experimental groups). Animals were exposed to their respective environments (CE or EE) for 60 consecutive days (from PND 60 to PND 120). At four months, animals underwent the following behavioral tests: PT, OFT, ST, TST, and rotarod. Twenty-four hours after the last behavioral test, a subgroup of animals was weighed and euthanized by cardiac perfusion to evaluate adult hippocampal neurogenesis. Another subgroup of animals was euthanized by cervical dislocation, their brains were removed, and the hippocampus, prefrontal cortex, and striatum were dissected to measure monoamine levels.

### 4.4. Behavioral Tests

All tests were performed during the light phase of the day–night cycle under low-intensity light (12 lux). All behavior tests were recorded by a video camera (HD Pro Webcam C920 Logitech, Newark, CA, USA) positioned above the apparatuses and were analyzed by an experimenter blind to the experimental groups. The apparatuses were cleaned with 10% ethanol between animals to avoid odor cues.

#### 4.4.1. Locomotor Activity and Motor Function

(a)Open field test (OFT): Locomotor activity was assessed in the open field apparatus, which consisted of a wooden box measuring 40 × 60 × 52 cm. The floor was divided into central and periphery regions, where activity in the center measures vs. the periphery of the apparatus is considered a proxy for anxiety-like behavior [92]. The distance traveled and the time spent in the center of the arena (13.5 × 30 cm) were measured for 6 min. Data were analyzed using the ANY-maze video-tracking system (Stoelting Co., Wood Dale, IL, USA).(b)Pole test (PT): The motor function was assessed by the PT as previously described [93] with some modifications. The testing apparatus consisted of a vertical rough-surfaced pole (54 cm height, 1 cm diameter), and the mouse was placed head-upward on top. The time between orienting downward and descending to the floor (time to descend) was measured. Each animal completed five trials, and the best performance was used. If the mouse could not turn entirely downward, fell, or slipped down, the default time of 120 s was recorded and taken as the maximal severity of impairment.(c)Accelerating rotarod test: The motor performance on the rotarod (Insight^®^, São Paulo, Brazil) was assessed as previously described [94,95] with some modifications. For the accelerating task, mice were once acclimatized to the apparatus for 120 s at a constant speed of 5 rpm. After a 120 min resting interval, each animal received 4 trials, with a progressively increasing speed, from 5 to 37 rpm over 5 min, with a 60 min interval between each trial. The time that each animal stayed on the rod before falling (the latency time for the first fall) and the number of falls were measured. Results were expressed as the average of the four sessions.

#### 4.4.2. Depressive-like Behavior Analyses

(a)Tail suspension test (TST): The TST was initially proposed in [96], and it is based on the evaluation of immobility in a situation of “behavioral despair”. Briefly, mice were suspended by their tail about 50 cm above the floor using adhesive tape. Mice were considered immobile only when they hung passively and were completely motionless. Immobility time was assessed during a 6 min period for each mouse.(b)Splash test (ST): The ST was performed using a previously described protocol [97] with some modifications. Animals were individually placed in a clear cylinder for 5 min for acclimatization to the apparatus. After habituation, a 10% sucrose solution was squirted onto the dorsal coat of each mouse. This procedure usually induces grooming behaviors because of its viscosity and palatable flavor. After applying sucrose solution, the latency to initiate grooming and the time spent grooming were recorded for 5 min as an index of self-care and motivational behavior.

### 4.5. Measurement of Brain Monoamine Levels

Twenty-four hours after behavioral testing, animals were quickly decapitated, their brains were removed, and the hippocampus, striatum, and prefrontal cortex were dissected on ice, immediately frozen in liquid nitrogen, and stored at −80 °C until analysis. Norepinephrine (NE), dopamine (DA), and serotonin (5-HT) levels were determined by high-performance liquid chromatography (HPLC) with fluorometric detection [98]. Brain tissue was homogenized on ice in microtubes containing 0.2 M perchloric acid with 3 mM cysteine at 1:6 (*w*/*v*). The homogenate was centrifuged (12,000× *g*, 10 min, 4 °C), and the resulting supernatant (20 μL) was injected into the HPLC.

A standard curve was created with concentrations ranging from 0.016 to 2.50 ng/μL to estimate DA, 5-HT, and NE levels. HPLC-EC analysis was performed with a Jasco LC-2000 Plus System (Jasco, Hachioji, Tokyo, Japan) using an ACE^®^ C18 Ultra-Inert column at a flow rate of 0.6 mL/min and held at a temperature of 35 °C. Monoamines were eluted in an isocratic solution of acetate (12 mM acetic acid, 0.26 mM ethylenediamine tetraacetic acid)/methanol (86:14, *v*/*v*). The fluorescence was monitored using excitation at 279 nm and emission at 320 nm. The analyte peaks were identified by comparing the retention times of the respective standards. The peak areas were integrated to quantify samples by linear regression of the calibration curve. Values were expressed as pg/μL.

### 4.6. Tissue Processing for Immunohistochemistry

Twenty-four hours after behavioral testing, animals were deeply anesthetized with an intraperitoneal (i.p.) injection of xylazine (8 mg/kg) and ketamine (100 mg/kg) and transcardially perfused with 0.9% sodium chloride (NaCl) followed by 4% paraformaldehyde (PFA). The brains were removed, left in 4% PFA overnight at 4 °C, and then transferred to 30% sucrose. Following saturation in sucrose, serial coronal sections were obtained on a vibratome (Vibratome^®^, Series 1000, St. Louis, MO, USA) at 30 μm thickness. Sections were collected into a 1/6 section sampling fraction and stored in 0.1 M phosphate-buffered saline (PBS) with 0.5% sodium azide at 4 °C.

#### 4.6.1. Hippocampal Cell Proliferation and Neuronal Differentiation

Immunohistochemistry was performed against the endogenous cell cycle protein Ki-67 to evaluate hippocampal cell proliferation. Ki-67 is a mitotic marker expressed during all active phases of the cell cycle (G1, S, G2, and M) [99]. Immunohistochemistry against doublecortin (DCX) was performed to evaluate neuronal commitment and maturation. DCX is a microtubule-associated protein specifically expressed in migrating neuroblasts (i.e., immature neurons) [34].

The following primary antibodies were used for immunostaining: polyclonal rabbit Ki-67 (1:500; Vector Laboratories, Burlingame, CA, USA) and goat polyclonal primary antibody against DCX (1:400; Santa Cruz Biotechnology, Dallas, TX, USA). All immunohistochemistry procedures were performed with floating sections. Sections were incubated twice in 10 mM citric acid (dissolved in 0.1 M PBS, pH = 6.0) for 5 min at 95 °C to expose the nuclear antigens completely. After rinsing in 0.1 M PBS, sections were quenched in 3% H_2_O_2_ and 10% methanol in 0.1 M PBS for 10 min at RT. Following thorough rinsing in 0.1 M PBS, tissue was blocked with 5% blocking solution (5% normal goat serum in 0.1 M PBS with 0.25% Triton X-100) for 1 h at RT and incubated with primary antibody diluted in 1% normal goat serum and 0.1% Triton X-100 in PBS at 4 °C for 48 h. For all immunohistochemistry procedures, bound antibodies were visualized using an avidin-biotin-peroxidase complex system (Vectastain ABC Elite Kit, Vector Laboratories) with 3,3′-diaminobenzidine (DAB; Vector Laboratories) as a chromogenic substrate. The sections were mounted onto 2% gelatin-coated microscope glass slides, dehydrated in a graded series of increasing ethanol and xylene solutions, and finally coverslipped with water-free mounting medium Entellan (Merk, Darmstadt, Germany).

#### 4.6.2. Morphological Quantification

All morphological analyses were performed on coded slides, with the experimenter blinded to the identity of the samples. The total numbers of Ki-67- and DCX-immunopositive cells present in the subgranular zone (SGZ) of the entire dentate gyrus (DG) (from 1.34 to 3.52 posterior to Bregma), the dorsal DG (from 1.34 to 2.30 posterior to Bregma; approximately six coronal sections per series/brain), and the ventral DG (from 2.30 to 3.52 posterior to Bregma; approximately six coronal sections per series/brain) [100] were quantified by manually counting all immunopositive cells present within 2–3 cell diameters of the SGZ, using an Olympus microscope (Olympus BX41, Center Valley, PA, USA) equipped with ×20 and ×40 objectives. Results were expressed as the total number of labeled cells in the DG sub-region of the hippocampus by multiplying the average number of labeled cells per DG section by the total number of 30 μm thick sections containing the entire DG SGZ (73 sections), the dorsal DG (nearly 36 sections), or the ventral DG (nearly 36 sections). Images were processed with Microsoft PowerPoint 2010. Only contrast enhancements and color level adjustments were made.

#### 4.6.3. Arborization Analysis

Dendritic arborization of immature neurons labeled with DCX was performed using Sholl analysis [88]. The slides were scanned using the ZEISS Axio Scan.Z1 slide scanner with a 20× objective and the ZEN Wildfield 2012 Blue Edition software (https://www.zeiss.com/microscopy/en/products/software/zeiss-zen.html). In summary, five sections containing the dorsal region and five sections containing the ventral region of the hippocampal DG were randomly selected from each animal. In each section, the DCX-positive cell with the least overlap with other cells and the greater number of visible dendritic branches was chosen for further analysis. This process resulted in five traces of DCX-positive cells from the dorsal hippocampus and five from the ventral hippocampus for each animal, resulting in 60 cells per experimental group. The dendrite length and the number of intersections per neuron were evaluated using Fiji software (https://fiji.sc/#download) [101,102,103]. The slides and images were coded to ensure the experimenter was blind to the experimental conditions.

### 4.7. Statistical Analysis

Statistical analyses were performed using the software Statistica 13.3 (StatSoft Inc., Tulsa, OK, USA). Normality was evaluated with the Kolmogorov–Smirnov test. All data are presented as mean ± SEM. The effects of genotype and environment were analyzed using a two-way analysis of variance (ANOVA), except for the number of dendritic intersections per ring, which was analyzed using a two-way repeated-measure ANOVA. When appropriate, the ANOVA was followed by the Bonferroni post hoc test. Correlation coefficients (*r2*) for the relationship between monoamine levels and motor performance (latency to fall from rotarod) were analyzed by estimating Spearman’s correlation coefficients (*r*). For all analyses, *p* values ≤ 0.05 were considered statistically significant.

## 5. Conclusions

Although the genetic cause of HD has been known since 1993, no cure or disease-modifying treatments are available. While pharmacological approaches targeting DNA and RNA show incredible promise for treating this genetic disorder, technical challenges and limitations must be overcome. Here, we show that EE, a non-pharmacological intervention, can positively impact some aspects of neuroplasticity and behavior in the YAC128 HD mouse model. EE prevented weight gain, depressive-like behaviors, and motor deficits in young YAC128 HD mice. In addition, EE affected monoamine levels in the striatum and the number of new cells in the hippocampus. These neuroplastic alterations did not negatively impact the behavioral parameters analyzed, unlike the pharmacological options currently available for HD, which are known to have a negative effect on mood [22,23]. These findings support prior studies and emphasize the potential of non-pharmacological interventions like EE in delaying disease progression and enhancing the quality of life for HD patients. This approach becomes crucial as we await further advancements in gene therapy. By exploring alternative avenues, we can better manage HD and offer hope to patients and their families. However, continued research is vital to fully understand the potential benefits of EE and how this can be incorporated into current HD treatment strategies. In conclusion, our study highlights the importance of non-pharmacological interventions as complementary approaches in HD management. By pursuing alternative avenues and improving our understanding of the benefits of EE, we can better support HD patients and work towards future breakthroughs in treatment.

## Figures and Tables

**Figure 1 ijms-24-12607-f001:**
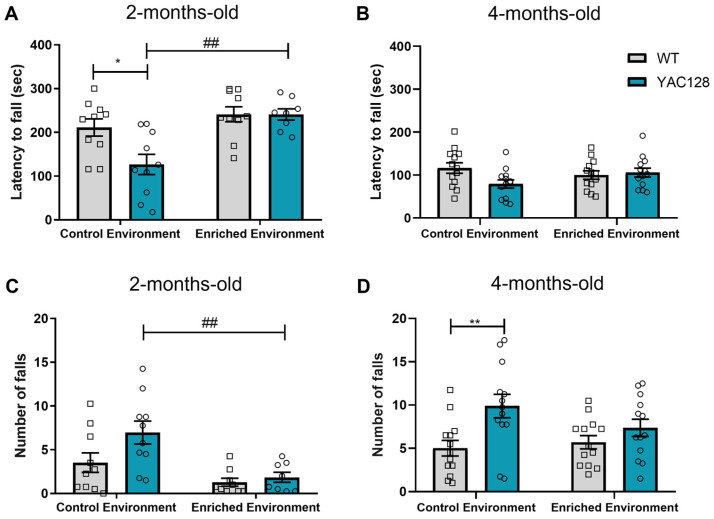
Effects of the genotype and environment on motor performance in WT and YAC128 mice as assessed with the rotarod test. Latency to fall (**A**,**B**) and the number of falls (**C**,**D**) in the rotarod test for 2- (**A**,**C**) and 4-month-old WT and YAC128 mice (**B**,**D**). N = 8–13 (3–7 females/group and 5–8 males/group). * *p* < 0.05 and ** *p* < 0.01, represent WT versus YAC128 differences in the same environment. ## *p* < 0.01, represents CE versus EE differences within the same genotype.

**Figure 2 ijms-24-12607-f002:**
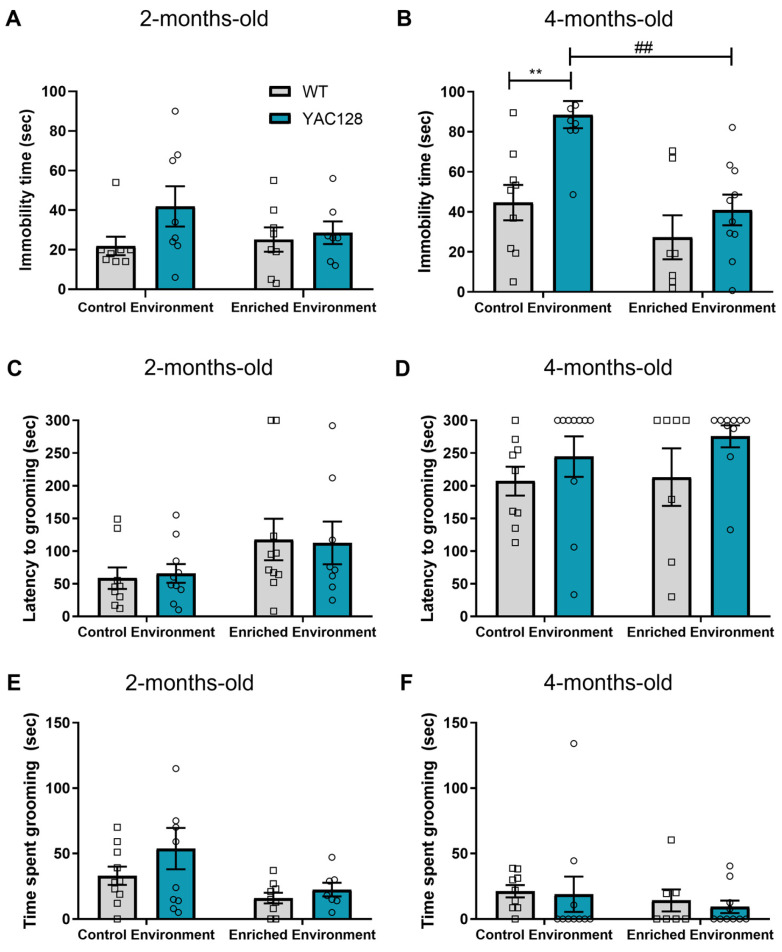
Effects of EE on depressive-like behaviors in WT and YAC128 mice were assessed with the tail suspension and splash tests. Immobility time in the tail suspension test (**A**,**B**), latency to grooming in the splash test (**C**,**D**), and time spent grooming in the splash test (**E**,**F**) were assessed for (**A**,**C**,**E**) 2- and (**B**,**D**,**F**) 4-month-old WT and YAC128 mice. N = 7–10 (2–5 females/group and 3–6 males/group). ** *p* < 0.01, represents WT versus YAC128 differences in the same environment. ## *p* < 0.01, represents CE versus EE differences within the same genotype.

**Figure 3 ijms-24-12607-f003:**
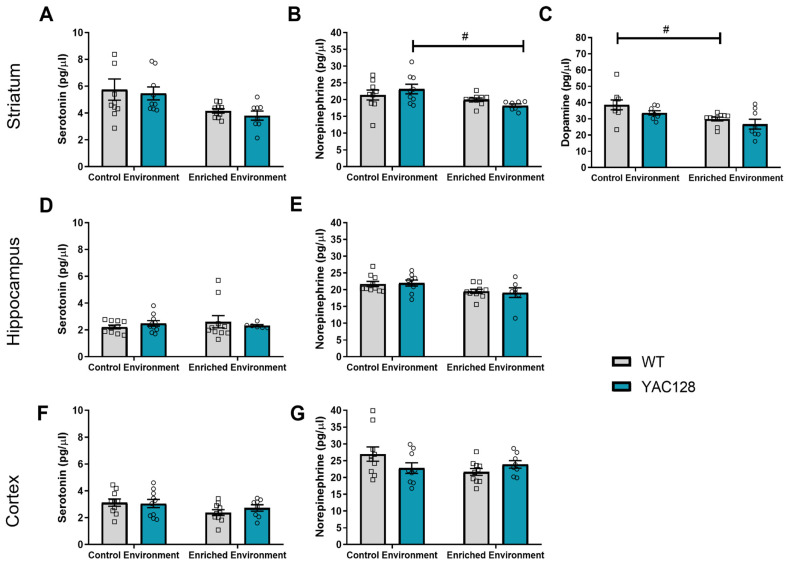
Monoamine levels in the striatum, hippocampus, and prefrontal cortex of WT and YAC128 mice in the CE and EE groups at 2 months of age. In the striatum, levels of 5-HT (**A**), NE (**B**), and DA (**C**) were analyzed by HPLC. EE was shown to decrease NE and DA striatal levels. In the hippocampus, no differences were found regarding 5-HT (**D**) and NE (**E**) levels. In the prefrontal cortex, no differences were found regarding 5-HT (**F**) and NE (**G**) levels. DA levels were not analyzed in the prefrontal cortex and hippocampus. Values are presented as pg/μL tissue ± SEM. N = 7–10 (1–5 females/group and 4–5 males/group). # *p* < 0.05, represents CE versus EE differences within the same genotype.

**Figure 4 ijms-24-12607-f004:**
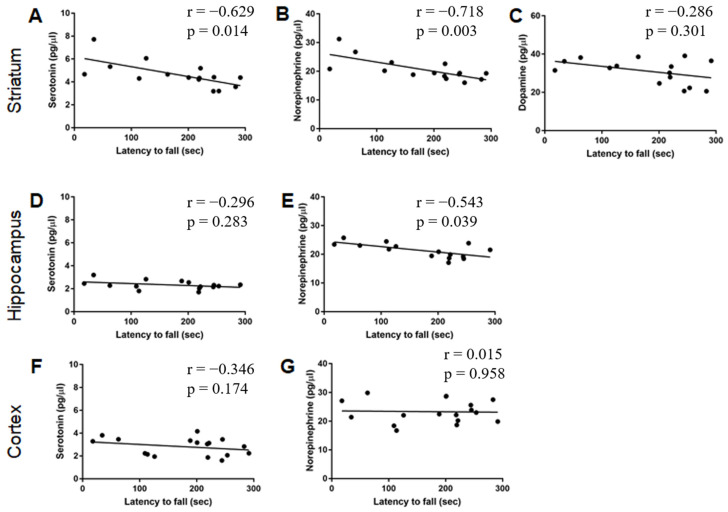
Correlation between monoamine levels (*y*-axis) and latency to fall in the rotarod (*x*-axis) in 2-month-old WT and YAC128 mice. The graphs represent correlations between monoamine levels and performance in the rotarod test in the striatum (**A**–**C**), hippocampus (**D**,**E**), and prefrontal cortex (**F**,**G**). The *Spearman* correlation test showed a significant correlation in (**A**,**B**,**E**). N of pairs = 15–17.

**Figure 5 ijms-24-12607-f005:**
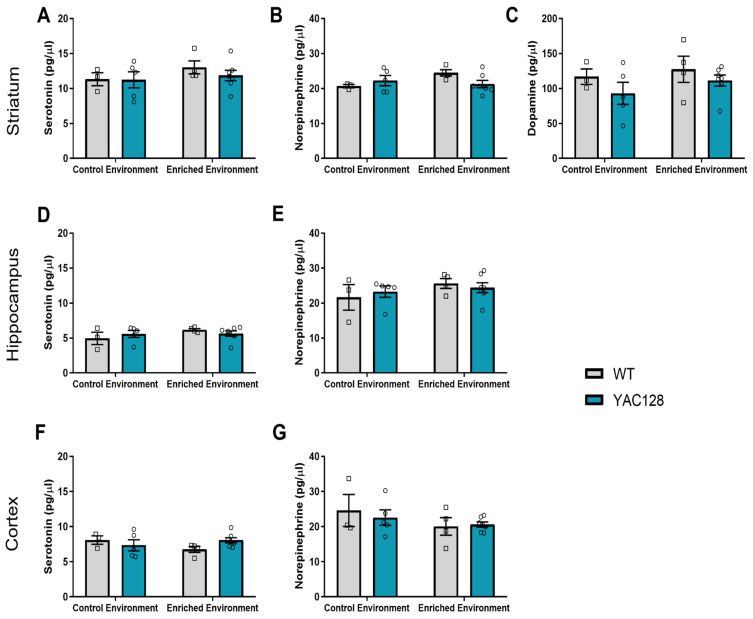
Monoamine levels in the striatum (**A**–**C**), hippocampus (**D**,**E**), and prefrontal cortex (**F**,**G**) of WT and YAC128 mice in the CE and EE groups at 4 months of age. Values are presented as pg/μL tissue. N = 3–7 (1–3 females/group and 2–5 males/group).

**Figure 6 ijms-24-12607-f006:**
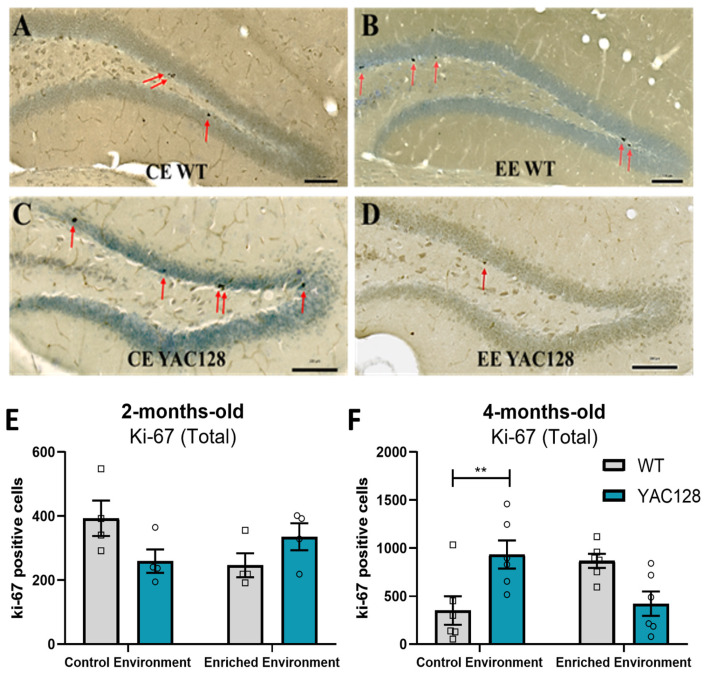
Hippocampal cell proliferation in 2- and 4-month-old animals. Evaluation of the effects of EE on DG cell proliferation in WT and YAC128 mice using the endogenous Ki-67 marker. Representative sections of the DG processed for Ki-67 in WT (**A**,**B**) and YAC128 (**C**,**D**) mice at 4 months of age. Arrows indicate Ki-67-positive cells. Quantification of the total number of Ki-67 positive cells in the DG of (**E**) 2- and (**F**) 4-month-old YAC128 and WT mice. Data are presented as means ± SEM. Scale bars = 100 μm. N = 4–6 (3–4 females/group and 2–3 males/group at 4-month-old). ** *p* < 0.01, represents WT versus YAC128 differences in the same environment.

**Figure 7 ijms-24-12607-f007:**
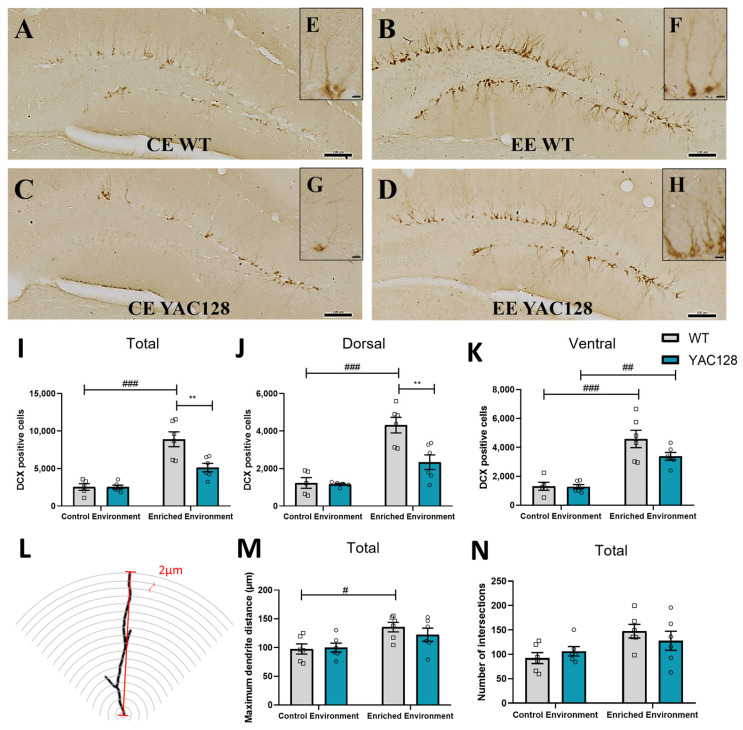
Effects of environmental enrichment on hippocampal neuronal differentiation (DCX) in 4-month-old animals. Representative sections of the DG processed for DCX immunohistochemistry in WT (**A**,**B**) and YAC128 (**C**,**D**) mice at 4 months of age. (**E**–**H**) represent magnified images showing individual neurons. Number of DCX-positive cells in the entire (**I**), dorsal (**J**), and ventral (**K**) SGZ of the hippocampal DG. (**L**) Representative image of a dendritic branch analysis with Sholl analysis. (**M**) represents the maximum distance from the soma to the end of the dendrites, and (**N**) represents the number of intersections per radius. Data are presented as means ± SEM. N = 5–6 (3–4 females/group and 2–3 males/group). Scale bars = 100 μm. ** *p* < 0.01, represents WT versus YAC128 differences in the same environment. # *p* < 0.05, ## *p* < 0.01 and ### *p* < 0.001, represent CE versus EE differences within the same genotype.

**Figure 8 ijms-24-12607-f008:**
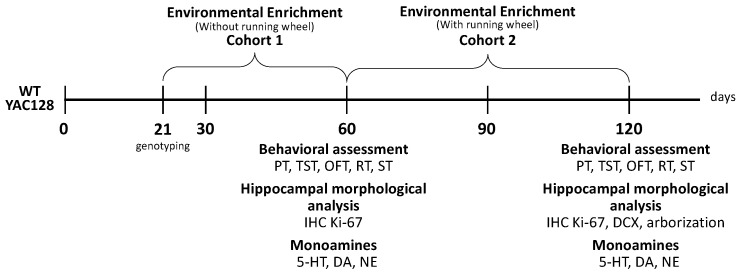
Experimental protocol. Cohort 1: Twenty-one-day WT and YAC128 female and male mice were housed in CE or EE conditions for 40 days. At PND 61, a battery of behavioral tests was applied: PT, OFT, ST, TST, and the rotarod test. At PND 65, mice were euthanized, their brains were removed, and the prefrontal cortex, striatum, and hippocampus were dissected to determine monoamine levels using high-performance liquid chromatography (HPLC). Another subgroup of animals was perfused, and their brains were collected for immunohistochemistry. Cohort 2: Sixty-day-old WT and YAC128 female and male mice were housed in CE or EE conditions for 60 days. At PND 120, a battery of behavioral tests was applied: PT, OFT, ST, TST, and the rotarod test. At PND 125, mice were euthanized, and their brains were removed. Some underwent immunohistochemistry, while others had the prefrontal cortex, striatum, and hippocampus dissected to determine monoamine levels using HPLC. PND = postnatal day; CE = control environment; EE = enriched environment; WT = wild-type; YAC = yeast artificial chromosome; 5-HT = serotonin; DA = dopamine; NE = norepinephrine.

**Table 1 ijms-24-12607-t001:** Body weight of WT and YAC128 mice at 2 and 4 months of age.

Age	Environment	Genotype	Body Weight (g)	N
2 months	Control	WT	22.3 ± 0.47	10 (5F–5M)
		YAC128	25.5 ± 1.07 *	10 (5F–5M)
	Enriched Environment	WT	23 ± 0.67	10 (5F–5M)
		YAC128	23.4 ± 0.62	08 (3F–5M)
4 months	Control	WT	26.78 ± 0.74	09 (4F–5M)
		YAC128	26.9 ± 1.00	10 (6F–4M)
	Enriched Environment	WT	25.29 ± 0.68	07 (4F–3M)
		YAC128	27.90 ± 0.74	10 (4F–5M)

Data represent mean ± SEM. * *p* < 0.05 versus WT, control group. F = females; M = males.

## Data Availability

The data supporting this study’s findings are available on request from the corresponding author.

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
