# Peer review of "Beyond Motor Deficits: Environmental Enrichment Mitigates Huntington’s Disease Effects in YAC128 Mice"

_ijms, 2023, doi:10.3390/ijms241612607_

Round 1
Reviewer 1 Report
In the present studies, the authors demonstrated that EE, a non-pharmacological intervention, can positively impact some aspects of neuroplasticity and behavior in the YAC128 HD mouse model. More specifically, they have showed that EE can prevent weight gain, the occurrence of depressive-like behaviors, and the development of motor deficits in 2- and 4-months old YAC128 HD mice. In addition, EE also had an effect on monoamine levels in the striatum and the number of new cells in the hippocampus. These neuroplastic alterations did not negatively impact the behavioural parameters analyzed, unlike the pharmacological options currently available for HD, which are known to have a negative effect on mood. Collectively, the present findings highlight the potential of non-pharmacological environmental interventions such as EE in delaying disease progression and improving the quality of life of patients afflicted with HD, particularly while we wait for further developments in gene therapy.
The manuscript was well-written and the conclusion was supported by the experimental data. I have some minor considerations:
1. Although, the table one showed the numbers of mice used, it is not clear how many male and female mice were used in each group. Please put this numbers from Figure 2 to Figure 8.
2. Any results showed difference in male and female mice?
Author Response
Reviewer #1
General Comments: In the present studies, the authors demonstrated that EE, a non-pharmacological intervention, can positively impact some aspects of neuroplasticity and behavior in the YAC128 HD mouse model. More specifically, they have showed that EE can prevent weight gain, the occurrence of depressive-like behaviors, and the development of motor deficits in 2- and 4-months old YAC128 HD mice. In addition, EE also had an effect on monoamine levels in the striatum and the number of new cells in the hippocampus. These neuroplastic alterations did not negatively impact the behavioural parameters analyzed, unlike the pharmacological options currently available for HD, which are known to have a negative effect on mood. Collectively, the present findings highlight the potential of non-pharmacological environmental interventions such as EE in delaying disease progression and improving the quality of life of patients afflicted with HD, particularly while we wait for further developments in gene therapy.
The manuscript was well-written and the conclusion was supported by the experimental data. I have some minor considerations:
Minor Comments:
- Although, the table one showed the numbers of mice used, it is not clear how many male and female mice were used in each group. Please put this numbers from Figure 2 to Figure 8.
R: We thank the reviewer for this observation. Please note that we have now included the number of males and females to the figures.
- Any results showed difference in male and female mice?
R: We did not find any statistically significant differences between males and females and for this reason, the data were analyzed together.

Reviewer 2 Report
A very good paper, however the discussions are little verbose and quite difficult to follow. A crisper enumeration would provide a better understanding of the results, maybe in bulleted form...
Otherwise no discernible flaws.
Author Response
Reviewer #2
- A very good paper, however the discussions are little verbose and quite difficult to follow. A crisper enumeration would provide a better understanding of the results, maybe in bulleted form...
R: We thank the reviewer for this comment. Please note that we have reworded part of the discussion so as to improve its readability. In addition, we have also divided this section into subsections, as suggested by the reviewer.

Reviewer 3 Report
22 July 2023
The review on the manuscript, titled “ENVIRONMENTAL ENRICHMENT PREVENTS MOTOR AND NON-MOTOR SYMPTOMS IN THE HUNTINGTON’S DISEASE YAC128 MOUSE MODEL” by Plácido E et al., submitted to International Journal of Molecular Sciences (IJMS)
Manuscript ID: ijms-2510581
Dear Authors,
Determining the optimal timing, duration, and type of environmental enrichment interventions, in addition to the need for more effective pharmacological treatments, is one of the current challenges. In the present research article titled ‘Environmental enrichment prevents motor and non-motor symptoms in the Huntington’s disease YAC128 mouse model’, Plácido and colleagues explore the potential benefits of environmental enrichment (EE) in preventing motor and non-motor symptoms in the Huntington's disease (HD) YAC128 mouse model. The manuscript investigated the effects of EE on body weight, depressive-like behaviors, anxiety-like behaviors, and motor performance in YAC128 mice. EE prevented weight gain, depressive-like behaviors, and motor deficits in the mice. It also influenced neurotransmitter levels in the striatum and hippocampus, mediating beneficial effects without negatively affecting observed behavioral parameters, differentiating it from some pharmacological treatments for HD. EE highlights the potential of non-pharmacological interventions to improve HD patients' quality of life and complement future treatments.
The primary strength of this manuscript is that it presents a comprehensive analysis of the effects of environmental enrichment on the prevention of motor and non-motor symptoms in the Huntington's disease YAC128 mouse model, using a variety of behavioral, morphological, and genetic assessments.
In general, I think the idea of this article is really interesting, and the authors’ fascinating observations on this timely topic may be of interest to the readers of IJMS. However, some comments, as well as some crucial evidence that should be included to support the author’s argumentation, needed to be addressed to improve the quality of the manuscript, its adequacy, and its readability prior to its publication in the present form. My overall opinion is to publish this research article after the author has carefully considered my comments and suggestions below.
Please consider the following comments:
1. I recommend revising the title. In its current form, I find it to be overly verbose, unclear, and insufficiently specific. Indeed, it could be reworded to increase readability and clarity. Suggestions: "Unlocking Hope: Environmental Enrichment Rescues Huntington's Disease YAC128 Mice from Motor and Non-Motor Symptoms"; "Beyond Motor Deficits: How Environmental Enrichment Shields YAC128 Mice from Huntington's Disease Effects" [1–3].
2. A graphical abstract that will visually summarize the main findings of the manuscript is highly recommended.
3. Abstract: I would like the authors to make as much effort for this section as for the rest of the manuscript. Please present the abstract in 200 words (preferably 200–220 words, max. 250) according to the guidelines of the journal [4], focusing on proportionally presenting the background, methods, results, and conclusion (without the headings of subsections). The background should include the general background (one to two sentences), the specific background (two to three sentences), and "the current issue addressed to this study" (one sentence), leading to the objectives. In this subsection, I would like the authors to lay out basic information, a problem statement, and their motivation to break off. The methods should clarify the authors’ approach, such as study design and variables, to solving the problem and/or making progress on the problem. The results should close with a single sentence putting the results in a more general context. The conclusion should open with one sentence describing the main result using such words as “Here we show”, which should be followed by statements such as the potential and the advance this study has provided in the field, and finally a broader perspective (two to three sentences) readily comprehensible to a scientist in any discipline [5–8].
4. Keywords: Please list ten keywords chosen from Medical Subject Headings (MeSH) [9] and use as many as possible in the title and in the first two sentences of the abstract [7,8]. I would suggest adding “Huntington's disease” and “Neuronal Plasticity” as keywords.
5. Introduction: The authors need to fully reorganize this section with several paragraphs made up of about 1000 words, introducing information on the main constructs of this protocol, which should be understood by a reader in any discipline, and making it persuasive enough to put forward the main purpose of the current research the author has conducted and the specific purpose the author has intended by this protocol. I would like to encourage the authors to present the introduction starting with the general background, proceeding to the specific background, rationales, and finally the current issue addressed to this study, leading to the objectives. Those main structures should be organized in a logical and cohesive manner [10].
6. In this regard, the manuscript would greatly benefit from incorporating a discussion about the underlying neural substrates that could be involved in the observed effects of EE. By delving into the neural mechanisms, the manuscript can provide deeper insights into the neurobiological basis of HD and how EE might modulate disease progression. Also, I would suggest the authors discuss the neuroanatomical and neurochemical alterations seen in HD, particularly in the YAC128 mouse model, and how these changes could contribute to motor and non-motor symptoms. It would be beneficial to elaborate on the regions of the brain affected in HD, such as the striatum and hippocampus, and how they are related to the observed behaviors in the mice, as well as emphasize the role of neurotransmitters, such as dopamine, norepinephrine, and serotonin, in HD pathology and their potential modulation by EE [11,12]. Discuss how changes in monoamine levels may influence motor function, mood, and cognition in HD and how environmental factors could interact with these pathways. By incorporating discussions on the neural substrates, the manuscript can provide a more comprehensive understanding of the potential mechanisms underlying the effects of EE in HD and enrich the overall scientific context of the study [13–15]. In addition, the following works, but not limited to, may enhance the value of this manuscript [16–19].
7. Methods: I recommend opening this section with a short introductory paragraph and citing more references to ensure the reliability and integrity of the evidence in the study design the authors built and the methodology they have decided to apply. This section should include more information about the number of animals in each group and the rationale for the group sizes (sample size calculation). Also, genotyping is mentioned in the methods section, but it might be helpful to provide more details on the technique used for genotyping the mice.
8. Results: In the results section, it would be beneficial to present the data in a more organized manner, such as using tables or graphs in colors. This will make it easier for readers to interpret the results. Furthermore, I suggest discussing the significance of the results in the context of previous literature. How do the findings compare with other studies that have investigated the effects of environmental enrichment in HD mouse models or other neurological disorders? I suggest closing this section with a paragraph that puts the results into a more general context.
9. Discussion: The discussion section lacks a clear and structured organization. I would like the authors to present this section by opening with an introductory paragraph, followed by a summary of the previous section. Then, I expect the authors to develop arguments clarifying the potential of this study as an extension of the previous work, the implication of the findings, how this study could facilitate future research, the ultimate goal, the challenge, the knowledge and technology necessary to achieve this goal, the statement about this field in general, and finally the importance of this line of research. It is particularly important to present its limits, its merits, and the potential translation of this protocol into clinical practice. Having that said, I suggest breaking it down into subsections that focus on specific aspects of the study, such as motor symptoms, non-motor symptoms, monoamine levels, and neuroplasticity effects. This will make it easier for readers to follow the flow of the discussion. Also, it would be helpful to compare and contrast the current findings with previous studies. Identify similarities and differences in the effects of environmental enrichment on motor and non-motor symptoms of HD.
10. Conclusion: I believe that presenting this section with 150–200 words would benefit from a single paragraph that presents some thoughtful and in-depth considerations by the authors as experts in order to convey the main message. The authors should make an effort to explain the theoretical implications as well as the translational application of their research. In order to understand the significance of this field, I believe it would be necessary to discuss theoretical and methodological avenues in need of refinement as well as future research directions [20,21].
11. Tables and Figures: According to the journal’s guidelines, please provide a short explanatory caption for the table within the text.
12. References: The authors should consider revising the bibliography, as there are several incorrect citations. Indeed, according to the journal’s guidelines [4], Please abbreviate the authors first name with a period, use a semicolon between authors, and list ten authors ending with et al. Also, please pay close attention to the journal’s name, year, volume, and page number.
Overall, the manuscript contains eight figures, a supplementary table, and 96 references. I believe that the manuscript may have merits in presenting its potential to inform future research on the use of EE as a non-pharmacological intervention for HD as well as its contribution to a broader understanding of the role of environmental factors in neurodegenerative diseases. I hope that, after careful revisions, the manuscript can meet the journal’s high standards for publication. I declare no conflict of interest regarding this manuscript.
Best regards,
Reviewer
References:
- https://plos.org/resource/how-to-write-a-great-title/
- https://www.nature.com/nature-index/news-blog/how-to-write-a-good-research-science-academic-paper-title
- https://www.indeed.com/career-advice/career-development/catchy-title
- https://www.mdpi.com/journal/ijms/instructions
- https://www.scribbr.com/dissertation/abstract/
- https://writing.wisc.edu/handbook/assignments/writing-an-abstract-for-your-research-paper/
- https://doi.org/10.5812/ijem.100159
- https://doi.org/10.4103/sja.SJA_685_18
- https://meshb.nlm.nih.gov/
- https://dept.writing.wisc.edu/wac/writing-an-introduction-for-a-scientific-paper/
- https://doi.org/10.3390/biomedicines11030945
- https://doi.org/10.1016/j.neubiorev.2023.105163
- https://doi.org/10.3390/biomedicines10122999
- https://doi.org/10.3389/fpsyt.2023.1225755
- https://www.frontiersin.org/articles/10.3389/fnins.2023.1098573/full
- https://doi.org/10.3390/brainsci12050676
- https://doi.org/10.1007/s00702-022-02513-5
- https://doi.org/10.1007/s43440-020-00067-5
- https://doi.org/10.3390/ijms222212499
- https://doi.org/10.3163/1536-5050.103.2.001
- https://www.scribbr.com/dissertation/discussion/
22 July 2023
The review on the manuscript, titled “ENVIRONMENTAL ENRICHMENT PREVENTS MOTOR AND NON-MOTOR SYMPTOMS IN THE HUNTINGTON’S DISEASE YAC128 MOUSE MODEL” by Plácido E et al., submitted to International Journal of Molecular Sciences (IJMS)
Manuscript ID: ijms-2510581
Dear Authors,
After evaluating the English proficiency, it has been determined that some minor revisions to the English language are necessary. While the overall communication is clear and understandable, certain areas could benefit from slight improvements in grammar, syntax, and word choice. Paying attention to detail, such as refining sentence structure and ensuring proper tense usage, will enhance the coherence and fluency of the written work as a whole. Making minor editing adjustments can lead to an improvement in English language proficiency.
Best regards,
Reviewer
Author Response
Reviewer #3
Specific Comments: Please consider the following comments:
- I recommend revising the title. In its current form, I find it to be overly verbose, unclear, and insufficiently specific. Indeed, it could be reworded to increase readability and clarity. Suggestions: "Unlocking Hope: Environmental Enrichment Rescues Huntington's Disease YAC128 Mice from Motor and Non-Motor Symptoms"; "Beyond Motor Deficits: How Environmental Enrichment Shields YAC128 Mice from Huntington's Disease Effects" [1–3].
R: We Thank the reviewer for this suggestion and have now changed the title as suggested.
- A graphical abstract that will visually summarize the main findings of the manuscript is highly recommended.
R: We Thank the reviewer for this suggestion and have now created a graphic abstract to accompany our manuscript. This has been included in our resubmission.
- Abstract: I would like the authors to make as much effort for this section as for the rest of the manuscript. Please present the abstract in 200 words (preferably 200–220 words, max. 250) according to the guidelines of the journal [4], focusing on proportionally presenting the background, methods, results, and conclusion (without the headings of subsections). The background should include the general background (one to two sentences), the specific background (two to three sentences), and "the current issue addressed to this study" (one sentence), leading to the objectives. In this subsection, I would like the authors to lay out basic information, a problem statement, and their motivation to break off. The methods should clarify the authors’ approach, such as study design and variables, to solving the problem and/or making progress on the problem. The results should close with a single sentence putting the results in a more general context. The conclusion should open with one sentence describing the main result using such words as “Here we show”, which should be followed by statements such as the potential and the advance this study has provided in the field, and finally a broader perspective (two to three sentences) readily comprehensible to a scientist in any discipline [5–8].
R: Please note that we have improved the Abstract as per the reviewer’s suggestion.
- Keywords: Please list ten keywords chosen from Medical Subject Headings (MeSH) [9] and use as many as possible in the title and in the first two sentences of the abstract [7,8]. I would suggest adding “Huntington's disease” and “Neuronal Plasticity” as keywords.
R: Please note that the two keywords suggested have now been added.
- Introduction: The authors need to fully reorganize this section with several paragraphs made up of about 1000 words, introducing information on the main constructs of this protocol, which should be understood by a reader in any discipline, and making it persuasive enough to put forward the main purpose of the current research the author has conducted and the specific purpose the author has intended by this protocol. I would like to encourage the authors to present the introduction starting with the general background, proceeding to the specific background, rationales, and finally the current issue addressed to this study, leading to the objectives. Those main structures should be organized in a logical and cohesive manner [10].
R: We thank the reviewer for their comments regarding the writing style employed throughout our manuscript. Indeed, the majority of the comments provided are primarily focused on our writing style, rather than the scientific merit of our study. We’d like to point that our writing style is in keeping with our numerous publications in this field (see for example refs. [1-4] below), and that the other two reviewers did not point any major issues with the current organization of our manuscript. As such, we have respectfully chosen to keep the overall organization, structure and writing style of our paper. Nevertheless, we have made some modifications throughout (including in the Introduction) following the reviewer’s suggestions and so as to improve its readability.
References:
- de Paula Nascimento-Castro C, Winkelmann-Duarte EC, Mancini G, Welter PG, Plácido E, Farina M, Gil-Mohapel J, Rodrigues ALS, de Bem AF, Brocardo PS. Temporal Characterization of Behavioral and Hippocampal Dysfunction in the YAC128 Mouse Model of Huntington's Disease. Biomedicines. 2022 Jun 17;10(6):1433. doi: 10.3390/biomedicines10061433.
- de Paula Nascimento-Castro C, Wink AC, da Fônseca VS, Bianco CD, Winkelmann-Duarte EC, Farina M, Rodrigues ALS, Gil-Mohapel J, de Bem AF, Brocardo PS. Antidepressant Effects of Probucol on Early-Symptomatic YAC128 Transgenic Mice for Huntington's Disease. Neural Plast. 2018 Aug 14;2018:4056383. doi: 10.1155/2018/4056383.
- Brocardo PS, McGinnis E, Christie BR, Gil-Mohapel J. Time-Course Analysis of Protein and Lipid Oxidation in the Brains of Yac128 Huntington's Disease Transgenic Mice. Rejuvenation Res. 2016 Apr;19(2):140-8. doi: 10.1089/rej.2015.1736.
- da Fonsêca VS, da Silva Colla AR, de Paula Nascimento-Castro C, Plácido E, Rosa JM, Farina M, Gil-Mohapel J, Rodrigues ALS, Brocardo PS. Brain-Derived Neurotrophic Factor Prevents Depressive-Like Behaviors in Early-Symptomatic YAC128 Huntington's Disease Mice. Mol Neurobiol. 2018 Sep;55(9):7201-7215. doi: 10.1007/s12035-018-0890-6.
- In this regard, the manuscript would greatly benefit from incorporating a discussion about the underlying neural substrates that could be involved in the observed effects of EE. By delving into the neural mechanisms, the manuscript can provide deeper insights into the neurobiological basis of HD and how EE might modulate disease progression. Also, I would suggest the authors discuss the neuroanatomical and neurochemical alterations seen in HD, particularly in the YAC128 mouse model, and how these changes could contribute to motor and non-motor symptoms. It would be beneficial to elaborate on the regions of the brain affected in HD, such as the striatum and hippocampus, and how they are related to the observed behaviors in the mice, as well as emphasize the role of neurotransmitters, such as dopamine, norepinephrine, and serotonin, in HD pathology and their potential modulation by EE [11,12]. Discuss how changes in monoamine levels may influence motor function, mood, and cognition in HD and how environmental factors could interact with these pathways. By incorporating discussions on the neural substrates, the manuscript can provide a more comprehensive understanding of the potential mechanisms underlying the effects of EE in HD and enrich the overall scientific context of the study [13–15]. In addition, the following works, but not limited to, may enhance the value of this manuscript [16–19].
R: We’d like to highlight that the points listed by the reviewer are indeed included in the Discussion Section. To avoid unnecessary repetition and ensure the length of manuscript remains acceptable, we have chosen to not repeat this information in the Introduction.
- Methods: I recommend opening this section with a short introductory paragraph and citing more references to ensure the reliability and integrity of the evidence in the study design the authors built and the methodology they have decided to apply. This section should include more information about the number of animals in each group and the rationale for the group sizes (sample size calculation). Also, genotyping is mentioned in the methods section, but it might be helpful to provide more details on the technique used for genotyping the mice.
R: We Thank the reviewer for raising these points. Please see that the total number of animals and the number of males and females used in the various experiments have now been included in the figure captions. In addition, we have also added more details to the genotyping protocol.
- Results: In the results section, it would be beneficial to present the data in a more organized manner, such as using tables or graphs in colors. This will make it easier for readers to interpret the results. Furthermore, I suggest discussing the significance of the results in the context of previous literature. How do the findings compare with other studies that have investigated the effects of environmental enrichment in HD mouse models or other neurological disorders? I suggest closing this section with a paragraph that puts the results into a more general context.
R: We Thank the reviewer for raising these points. Please note that we have now updated our Figures in colour now. In addition, an in keeping with the general structure and organization of our previously published manuscripts, we have chosen to discuss the significance of the results reported in this Section in the Discussion section.
- Discussion: The discussion section lacks a clear and structured organization. I would like the authors to present this section by opening with an introductory paragraph, followed by a summary of the previous section. Then, I expect the authors to develop arguments clarifying the potential of this study as an extension of the previous work, the implication of the findings, how this study could facilitate future research, the ultimate goal, the challenge, the knowledge and technology necessary to achieve this goal, the statement about this field in general, and finally the importance of this line of research. It is particularly important to present its limits, its merits, and the potential translation of this protocol into clinical practice. Having that said, I suggest breaking it down into subsections that focus on specific aspects of the study, such as motor symptoms, non-motor symptoms, monoamine levels, and neuroplasticity effects. This will make it easier for readers to follow the flow of the discussion. Also, it would be helpful to compare and contrast the current findings with previous studies. Identify similarities and differences in the effects of environmental enrichment on motor and non-motor symptoms of HD.
R: We have reorganized the Discussion Section as suggested by the reviewer and so as to increase its readability. In addition, we have also divided this section into subsections, as suggested by the reviewer. Finally, please note that we have also included a sentence regarding the potential translation of our protocol into clinical practice.
- Conclusion: I believe that presenting this section with 150–200 words would benefit from a single paragraph that presents some thoughtful and in-depth considerations by the authors as experts in order to convey the main message. The authors should make an effort to explain the theoretical implications as well as the translational application of their research. In order to understand the significance of this field, I believe it would be necessary to discuss theoretical and methodological avenues in need of refinement as well as future research directions [20,21].
R: We have made some adjustments to the Conclusion, so as to improve its readability.
- Tables and Figures: According to the journal’s guidelines, please provide a short explanatory caption for the table within the text.
R: Please note that the Table already includes a short caption, as per the Journal’s Guidelines.
- References: The authors should consider revising the bibliography, as there are several incorrect citations. Indeed, according to the journal’s guidelines [4], Please abbreviate the authors first name with a period, use a semicolon between authors, and list ten authors ending with et al. Also, please pay close attention to the journal’s name, year, volume, and page number.
R: We have now reviewed our References List to ensure all references are formatted and presented according to the Journal’s Guidelines.

Round 2
Reviewer 3 Report
4 August 2023
The 2nd review on the manuscript, titled “ENVIRONMENTAL ENRICHMENT PREVENTS MOTOR AND NON-MOTOR SYMPTOMS IN THE HUNTINGTON’S DISEASE YAC128 MOUSE MODEL” by Plácido E et al., submitted to International Journal of Molecular Sciences (IJMS)
Manuscript ID: ijms-2510581
Dear Authors,
In the present research article titled ‘Environmental enrichment prevents motor and non-motor symptoms in the Huntington’s disease YAC128 mouse model’, Plácido and colleagues explore the potential benefits of environmental enrichment (EE) in preventing motor and non-motor symptoms in the Huntington's disease (HD) YAC128 mouse model. I am pleased to see that the authors have attempted to revise the manuscript; nevertheless, the revisions remain partial in regard to my previous report. Prior to publication, I respectfully request that the authors consider my comments and revise the manuscript to meet the high standards of the journal. Please consider the following comments:
1. Keywords: Please list ten keywords chosen from Medical Subject Headings (MeSH) [9] and use as many as possible in the title and in the first two sentences of the abstract. I would suggest adding “Huntington's disease” and “Neuronal Plasticity” as keywords.
2. Introduction: Please clarify the objectives of this study in the last paragraph of the introduction, as suggested previously: The authors need to fully reorganize this section with several paragraphs made up of about 1000 words, introducing information on the main constructs of this protocol, which should be understood by a reader in any discipline, and making it persuasive enough to put forward the main purpose of the current research the author has conducted and the specific purpose the author has intended by this protocol. I would like to encourage the authors to present the introduction starting with the general background, proceeding to the specific background, rationales, and finally the current issue addressed to this study, leading to the objectives. Those main structures should be organized in a logical and cohesive manner.
3. Results: Please avoid writing statistical values in the text. I recommend presenting all statistical values in the tables. I suggest closing this section with a paragraph that puts the results into a more general context.
4. Discussion: Please refrain from using subsections in the discussion. A summary of the results should be described in a couple of paragraphs and please pay close attention to the following structure, as suggested previously: I would like the authors to present this section by opening with an introductory paragraph, followed by a summary of the previous section. Then, I expect the authors to develop arguments clarifying the potential of this study as an extension of the previous work, the implication of the findings, how this study could facilitate future research, the ultimate goal, the challenge, the knowledge and technology necessary to achieve this goal, the statement about this field in general, and finally the importance of this line of research. It is particularly important to present its limits, its merits, and the potential translation of this protocol into clinical practice. Having that said, I suggest breaking it down into subsections that focus on specific aspects of the study, such as motor symptoms, non-motor symptoms, monoamine levels, and neuroplasticity effects. This will make it easier for readers to follow the flow of the discussion. Also, it would be helpful to compare and contrast the current findings with previous studies. Identify similarities and differences in the effects of environmental enrichment on motor and non-motor symptoms of HD.
5. Methods: I recommend opening this section with a short introductory paragraph.
6. References: Please follow the journal’s guidelines. Page number should be punctuated with periods.
Overall, the manuscript contains eight figures, a supplementary table, and 103 references. I believe that the manuscript may have merits in presenting its potential to inform future research on the use of EE as a non-pharmacological intervention for HD as well as its contribution to a broader understanding of the role of environmental factors in neurodegenerative diseases. I hope that, after careful revisions, the manuscript can meet the journal’s high standards for publication. I declare no conflict of interest regarding this manuscript.
Best regards,
Reviewer
4 August 2023
The 2nd review on the manuscript, titled “ENVIRONMENTAL ENRICHMENT PREVENTS MOTOR AND NON-MOTOR SYMPTOMS IN THE HUNTINGTON’S DISEASE YAC128 MOUSE MODEL” by Plácido E et al., submitted to International Journal of Molecular Sciences (IJMS)
Manuscript ID: ijms-2510581
Dear Authors,
Based on the English proficiency assessment, it is noted that minor editing of the English language is required. While the overall communication is clear and understandable, there are some areas that could benefit from slight improvements in grammar, syntax, and word choice. Attention to detail, such as refining sentence structure and ensuring proper tense usage, will enhance the overall coherence and fluency of the written work. With some minor editing adjustments, the English language proficiency can be further enhanced.
Best regards,
Reviewer
Author Response
Response to the Reviewer 3
Manuscript ID: ijms-2510581
Dear Authors,
In the present research article titled ‘Environmental enrichment prevents motor and non-motor symptoms in the Huntington’s disease YAC128 mouse model’, Plácido and colleagues explore the potential benefits of environmental enrichment (EE) in preventing motor and non-motor symptoms in the Huntington's disease (HD) YAC128 mouse model. I am pleased to see that the authors have attempted to revise the manuscript; nevertheless, the revisions remain partial in regard to my previous report. Prior to publication, I respectfully request that the authors consider my comments and revise the manuscript to meet the high standards of the journal. Please consider the following comments:
- Keywords:Please list ten keywords chosen from Medical Subject Headings (MeSH) [9] and use as many as possible in the title and in the first two sentences of the abstract. I would suggest adding “Huntington's disease” and “Neuronal Plasticity” as keywords.
R: Please see that these 2 key-words were already added to the key-words, following their initial suggestion. We have respectfully maintained the remaining key-works as we believe they accurately capture main topics covered in our paper and are widely used in the field.
- Introduction: Please clarify the objectives of this study in the last paragraph of the introduction, as suggested previously: The authors need to fully reorganize this section with several paragraphs made up of about 1000 words, introducing information on the main constructs of this protocol, which should be understood by a reader in any discipline, and making it persuasive enough to put forward the main purpose of the current research the author has conducted and the specific purpose the author has intended by this protocol. I would like to encourage the authors to present the introduction starting with the general background, proceeding to the specific background, rationales, and finally the current issue addressed to this study, leading to the objectives. Those main structures should be organized in a logical and cohesive manner.
R: We respectfully disagree with the reviewer's assessment of the introduction's organization. It is important to note that we have already implemented several of the valuable suggestions provided earlier. Despite the differences in opinion, we believe that the revised Introduction serves its purpose and sets the stage for the rest of the paper.
- Results: Please avoid writing statistical values in the text. I recommend presenting all statistical values in the tables. I suggest closing this section with a paragraph that puts the results into a more general context.
R: We respectfully disagree with the reviewer's preference regarding how to present the STATS. It is important to recognize that the manner in which statistics are presented can be a subjective matter, subject to personal opinions and varying conventions in the field. While we value the reviewer's input and have carefully considered their perspective, we believe that the approach we have chosen for presenting the statistics aligns well with the standards commonly accepted within our research community. As with any aspect of academic work, there may be alternative ways to present data, and we acknowledge that individual preferences may differ. Our decision on the presentation of STATS is based on a thoughtful consideration of various factors and best practices in the relevant domain.
- Discussion: Please refrain from using subsections in the discussion. A summary of the results should be described in a couple of paragraphs and please pay close attention to the following structure, as suggested previously: I would like the authors to present this section by opening with an introductory paragraph, followed by a summary of the previous section. Then, I expect the authors to develop arguments clarifying the potential of this study as an extension of the previous work, the implication of the findings, how this study could facilitate future research, the ultimate goal, the challenge, the knowledge and technology necessary to achieve this goal, the statement about this field in general, and finally the importance of this line of research. It is particularly important to present its limits, its merits, and the potential translation of this protocol into clinical practice. Having that said, I suggest breaking it down into subsections that focus on specific aspects of the study, such as motor symptoms, non-motor symptoms, monoamine levels, and neuroplasticity effects. This will make it easier for readers to follow the flow of the discussion. Also, it would be helpful to compare and contrast the current findings with previous studies. Identify similarities and differences in the effects of environmental enrichment on motor and non-motor symptoms of HD.
R: We are very confused by the reviewer's comments on the discussion section. They've started the comment above by asking us to refrain from the use of subsections, while further in the comment they state “I suggest breaking it down into subsections that focus on specific aspects of the study”. In fact, the subsections were added to make the discussion easies to follow and to attend to a suggestion of Reviewer #2 (as well as the initial comment from Reviewer #3, which asked us to “break down [ the Discussion] into subsections that focus on specific aspects of the study”). We strongly believe that the discussion section greatly benefits from the inclusion of subsections, enhancing its clarity and organization. It is important to emphasize that we are not altering the content itself, but rather, we took into account the valuable feedback from Reviewer #2, who suggested the implementation of subsections to improve the overall flow and readability of the discussion. We firmly stand by the fact that the inclusion of subsections in the Discussion has been instrumental in making the content more accessible for the readers.
- Methods: I recommend opening this section with a short introductory paragraph.
R: We have chosen not to include that specific content in this section, as it primarily focuses on methods rather than serving as an introduction. It is essential to maintain the section's clarity and purpose, which is to comprehensively outline the methodology employed in our study. This is line with previous publications in this journal and in the field.
- References: Please follow the journal’s guidelines. Page number should be punctuated with periods.
R: We have attempted to correct any references that were not formatted according to the Journal’s guidelines. If we have missed any, these errors can be fixed during the proof corrections stage, in collaboration with the Journal editorial team.
The 2nd review on the manuscript, titled “ENVIRONMENTAL ENRICHMENT PREVENTS MOTOR AND NON-MOTOR SYMPTOMS IN THE HUNTINGTON’S DISEASE YAC128 MOUSE MODEL” by Plácido E et al., submitted to International Journal of Molecular Sciences (IJMS)
Manuscript ID: ijms-2510581
Dear Authors,
Based on the English proficiency assessment, it is noted that minor editing of the English language is required. While the overall communication is clear and understandable, there are some areas that could benefit from slight improvements in grammar, syntax, and word choice. Attention to detail, such as refining sentence structure and ensuring proper tense usage, will enhance the overall coherence and fluency of the written work. With some minor editing adjustments, the English language proficiency can be further enhanced.
R: Our Corresponding Author is a Canadian citizen (English speaker) and has personally reviewed the latest version of the manuscript to ensure the quality of the English language. Any minor grammatical or syntax errors can be fixed during the proof corrections stage.
